# The Flexural Strength Prediction of Carbon Fiber/Epoxy Composite Using Artificial Neural Network Approach

**DOI:** 10.3390/ma16155301

**Published:** 2023-07-28

**Authors:** Veena Phunpeng, Karunamit Saensuriwong, Thongchart Kerdphol, Pichitra Uangpairoj

**Affiliations:** 1School of Mechanical Engineering, Institute of Engineering, Suranaree University of Technology, 111 Maha Witthayalai Rd., Suranaree Sub-District, Mueang Nakhon Ratchasima District, Nakhon Ratchasima 30000, Thailand; karunamitsaen@gmail.com (K.S.); pichitrau@g.sut.ac.th (P.U.); 2Department of Electrical Engineering, Faculty of Engineering, Kasetsart University, 50 Ngamwongwan Rd., Chatuchak, Bangkok 10900, Thailand; thongchartkerd@gmail.com

**Keywords:** composite materials, carbon fiber/epoxy composite, artificial intelligence, artificial neural network, Levenberg–Marquardt backpropagation

## Abstract

There is a developing demand for natural resources because of the growing population. Alternative materials have been developed to address these shortages, concentrating on characteristics such as durability and lightness. By researching composite materials, natural materials can be replaced. It is vital to consider the mechanical properties of composite materials when selecting them for a specific application. This study aims to measure the flexural strength of carbon fiber/epoxy composites. However, the cost of forming these composites is relatively high, given the expense of composite materials. Consequently, this study seeks to reduce molding costs by predicting flexural strength. Conducting many tests for each case is costly; therefore, it is necessary to discover an economical method. To accomplish this, the flexural strength of carbon fiber/epoxy composites was investigated using an artificial neural network (ANN) technique to reduce the expense of material testing. The output parameter investigated was flexural strength, while input parameters included ply orientation, manufacturing, width, thickness, and graphite filler percentage. The scope alternative was determined by identifying the values of variables that substantially affect the flexural strength. The prediction of flexural strength was deemed acceptable if the mean squared error (MSE) value was less than 0.001, and the coefficient of determination (R^2^) was greater than or equal to 0.95. The obtained results demonstrated an MSE of 0.003039 and an R^2^ value of 0.95274, indicating a low prediction error and high prediction accuracy for all flexural strength data. Thus, the outcomes of this study provide accurate predictions of flexural strength in the composite materials.

## 1. Introduction

The increasing demand for natural materials can be attributed to the growth of construction, industry, and household devices. However, the utilization of natural materials has detrimental effects on the environment, and falls short in meeting human needs. It leads to negative consequences such as habitat degradation, biodiversity loss, desertification, and increased carbon dioxide levels in the atmosphere. To address these challenges, alternative materials have been developed to fulfill human requirements, while considering important material properties such as strength, weight, and usability [1,2,3,4]. Among the wide array of alternative materials, composite materials have gained significant popularity. They are composed of two or more distinct structures or chemical compositions that are mixed together. These materials must be visually distinguishable while exhibiting improved mechanical properties. Composite materials typically consist of a matrix material and reinforcing fibers [5,6,7,8,9,10]. Carbon fiber/epoxy composites, in particular, have exceptional mechanical properties, including high strength, fatigue resistance, corrosion resistance, and creep resistance [11,12,13]. These properties demonstrate the material’s ability to withstand external forces. In engineering applications, mechanical properties play a crucial role, serving as the primary consideration when selecting appropriate materials for a given task. Various testing methods, such as tensile, bending, and compression tests, are employed to determine these mechanical properties [14,15,16]. In this research, particular attention is given to flexural strength, which refers to a material’s ability to withstand a load or external force [17].

The flexural strength of carbon fiber/epoxy resin composites is influenced by various variables, including ply orientation, the number of layers, manufacturing process, and the percentage of graphite filler. Among these variables, ply orientation and the number of carbon fiber layers are considered as primary factors that significantly impact the flexural strength and other mechanical properties [18,19,20,21,22,23]. In a study conducted by M. Ghasemi et al. (2016) [18], polymer matrix composites (PMCs) using glass fiber/epoxy composites were subjected to thermal cycling, which involved varying temperature, ply orientation, volume of fiber, and the number of heating cycles. Their study revealed that optimal mechanical properties were attained when the ply orientation was at 0° and consisted of 8 layers in all cases. Furthermore, increasing the temperature enhanced the bonding between the fiber and the matrix, leading to improved mechanical properties. Another study by M. Ataeefard et al. (2014) [24] examined toner powder comprising a mixture of polystyrene and acrylic polymer using scanning electron microscopy (SEM) to analyze the size, shape, and cohesion of the polymer. Their research demonstrated that heating the toner resulted in improved binding between the two materials, particularly with the use of polystyrene acrylic. Moreover, the morphology analysis revealed that the incorporation of carbon particles in the toner enhanced the electrical conductivity of the material.

To determine the flexural strength when changing ply orientation and number of layers, multiple specimens must be tested for reliable results. However, this can lead to long production and testing times. Therefore, methods have been developed to reduce the time and materials required to form test specimens by predicting the flexural strength of materials. Nowadays, various innovative methods (such as artificial intelligence (AI), fuzzy) can be used to predict the interested things [25,26]. Artificial intelligence (AI) is a science of technology that has developed computer systems to perceive the process of thinking, acting, reasoning, adaptation, inference, and functioning of the human brain. AI is used in many lines (i.e., industry, surveying, and research) [27,28,29]. There are various methods for processing AI to accurately predict values of interest, and the most popular and suitable method for numerical analytics is the artificial neural network (ANN) [30].

The ANN method is an automated learning technique that mimics the operation of human neural networks (neurons). The ANN system is constructed by overlapping several layers and learning sample data, which are then used to detect patterns or classify the data so that the neural network system can process data as complexly as the human brain. Therefore, the ANN’s hidden layer must have multiple layers to process data sequentially, enabling it to compute more complex tasks [31,32,33,34,35]. For instance, M. Kazi et al. (2020) [36] presented an approach to using the ANN method to design cotton fiber-reinforced polymer composites with polypropylene to identify the amount of cotton fiber that affects the optimum mechanical properties. The input data included initiation energy, propagation energy, total energy, ductility index, tensile strength, modulus of elasticity, percentage elongation, and net weight of fiber, while the output data predicted the optimum filler content. Based on the results of the artificial intelligence method, the developed ANN could predict the optimum filler content of cotton fibers by considering the input mechanical properties. Recent research by S. A. Martini et al. (2023) [37] investigated the mechanical properties of recycled concrete aggregate (RCA). The researchers blended granulated blast-furnace slag and fly ash with RCA in various proportions (0, 20, 40, 60, and 100 percent). At varied time intervals (3 days, 7 days, and 28 days), uniaxial compressive and flexural tests were performed. Their study aimed to evaluate the impact of recycled aggregates using machine learning techniques. The findings demonstrated that the use of high levels of fly ash replacement (30%) combined with a high content of RCA (60% and 100%) resulted in higher compressive strength after 28 days compared to other mixtures. Regarding the equation developed to predict compressive strength at 3 days, it yielded satisfactory results, with an R^2^ value exceeding 0.80. Moreover, the proposed equation for predicting compressive strength at 28 days and 7 days proved to be highly accurate, with an R^2^ value above 0.90. These results highlight the potential of machine learning in analyzing and predicting the mechanical properties of recycled concrete aggregates. By leveraging such techniques, researchers and engineers can gain valuable insights and optimize the use of recycled materials in construction applications. A.R. Knowlton et al. (2021) [38] investigated composite materials using carbon nanotubes in combination with cement to develop composite materials. To obtain accurate and similar values before analysis, the data were validated and normalized with the number of hidden layers and neurons in each hidden layer to fit the data. They found that using two hidden layers and 20 neurons provided the most accurate prediction value. In a recent study conducted by Abushanab A. et al. (2023) [39], the flexural strength of reinforced concrete beams was evaluated using machine learning (ML) techniques. Various models, including SVM, DT, ADB, and GB, were employed for analysis. The dataset was randomly initialized, with 80% of the data used for training and the remaining 20% for testing. The GB model demonstrated the highest predictive accuracy among the tested models, as evidenced by the R^2^ values and the lowest maximum error. These results indicate that the GB model can effectively predict the flexural strength of reinforced concrete beams. Using machine learning, researchers can improve their understanding of the behavior of concrete beams and make precise predictions. This study’s findings contribute to advancing machine learning (ML)-based techniques in structural engineering, and offer valuable insights for optimizing the design and analysis of reinforced concrete structures.

Artificial neural network prediction has many algorithmic methods (i.e., Levenberg–Marquardt: LM, Bayesian regularization: BP, scaling conjugate gradient: SCG). The choice of algorithm depends on the extent of the data and the amount of data [40,41,42,43,44]. Jiaojiao F. (2018) [41] studied the estimated monthly mean daily global solar radiation using the neural network approach with the LM and BP algorithms. The results showed that both algorithms can predict the utilization of solar and heat energy, with a root mean squared error (RMSE) of 1.34 MJ·m^−2^. SCG is a method that is suitable for predicting information with a width of the dataset to reduce the scope before prediction. The LM algorithm is widely used for predicting mathematical data, and can reduce the range between mathematical data. Therefore, the LM algorithm is appropriately used in this research [42].

Building upon the existing literature, the primary objective of this study is to investigate the strength characteristics of carbon fiber/epoxy composites. It is worth noting that the production process of composite materials is relatively costly when compared to the expense of the materials themselves. Consequently, the focus of this research is to minimize the expenses associated with composite molding by accurately predicting the flexural strength. Conducting a significant number of tests for each case can be quite expensive. In order to mitigate the costs associated with material testing, the potential of artificial intelligence (AI) has been explored as a means of accurately predicting material strength. This research, in particular, concentrates on predicting the flexural strength derived from the 3-point bending test. By harnessing the power of AI to forecast flexural strength, the objective is to develop a valuable tool applicable to various scenarios, eliminating the necessity for new manufacturing processes, while still ensuring the desired level of flexural strength.

## 2. Methodology

### 2.1. Manufacturing

In this section, the manufacturing process of carbon fiber/epoxy composites is examined as a significant factor influencing their mechanical properties. The comparison is drawn explicitly between two methods: vacuum bagging and vacuum infusion. Additionally, the effect of adding graphite fillers to the matrix at various ratios is explored. By delving into these manufacturing aspects, a comprehensive understanding of how they impact the mechanical properties of carbon fiber/epoxy composites can be achieved.

#### 2.1.1. Materials

In this study, a 3 K carbon fiber fabric was employed, woven in a 1 × 1 plain weave pattern, with a weight of 200 g/m^2^ and a density of 1.8 g/cm^3^. The chosen matrix material is ER550, an epoxy resin, with a resin-to-hardener ratio of 100:35. Furthermore, a graphite filler with a particle size of 5 μm was incorporated into the composite material as a filler component.

#### 2.1.2. Preparation of Specimens

During the specimen preparation process, the resin and graphite filler are meticulously blended at varying ratios of 5 wt%, 7.5 wt%, 10 wt%, 12.5 wt%, and without any graphite filler to create the matrix. After reaching the desired ratios, a hardener is introduced into the mixture, enabling the manufacturing process of the carbon fiber/epoxy composite. Subsequently, the carbon fiber fabric is integrated with the resin–filler mixture to form the composite material. The specific manufacturing conditions for the carbon fiber/epoxy composite are detailed in Table 1. The manufacturing scope, as outlined in Table 1, consists of 103 specimens comprising 8 layers of fabric, which were adopted from the study by Phunpeng V. et al. [20]. The remaining 384 specimens were newly manufactured for this research. Figure 1 depicts an example of a specimen with eight layers of carbon fabric and graphite filler in a [0°/90°]_4s_ configuration. In Figure 2, a completed manufacturing sample is illustrated, showcasing various boundaries. For instance, the specimen labeled as “200_VB_5CF1_0%” in Figure 2 refers to the use of carbon fabric with a weight of 200 g/m^2^, produced through a Vacuum Bagging (VB) process. It comprises five layers of carbon cloth arranged in a [0°/90°/0°/90°/0°] configuration, with no graphite filler (0% content). The numbers 3, 20.98, and 1.2 correspond to the quantity of specimens, width, and thickness, respectively.

#### 2.1.3. Vacuum Bagging and Vacuum Infusion Processes

In this step, a vacuum bagging process is applied by laying down a carbon fiber cloth and adding a matrix (i.e., epoxy resin mixed with graphite filler and hardener) to each layer of carbon fiber fabric. After that, the process of sucking the air out with a pressure of −0.8 bar was applied, and the specimen was heated in an oven at 100 °C for 2 h.

The vacuum infusion process uses a manufacturing method similar to vacuum bagging process. Instead, a vacuum system is used to guide the matrix into the fabric layer and distribute it across the carbon fibers on the mold. The pressure is maintained at −0.8 bar for 15 min, after which the specimen is left to dry for 2 days, and then heated in an oven at 100 °C for 2 h. This would make it clearer and more concise [20]. An example of a specimen in the case of eight layers of carbon fabric with graphite filler in [0°/90°]_4s_ is shown in Figure 1 and Figure 2.

#### 2.1.4. Scanning Electron Microscopy (SEM)

In the study of carbon fiber/epoxy composites, the researchers examined the carbon fibers and graphite filler using scanning electron microscopy (SEM) spectroscopy. The carbon fibers and graphite filler were carefully positioned on a specimen stand, and then gold-plated through electron deposition for a duration of 2 min. Subsequently, the carbon fibers and graphite filler were subjected to SEM spectrophotometry to observe and analyze their respective characteristics. By progressively increasing the magnification, detailed images of the fiber and filler characteristics were captured, as depicted in Figure 1.

#### 2.1.5. Testing for Flexural Strength

The objective of this research was to evaluate the flexural strength using the 3-point bending test. The motivation for this study stems from its connection to previous research conducted on hybrid composites with waste graphite fillers for UAVs [20]. Considering the wings and other structural components of UAVs, it becomes evident that tensile and compression forces contribute significantly to bending. Hence, the focus on the bending test in this research is justified, as it helps assess the material’s ability to withstand bending stresses in relevant applications. The test was performed utilizing a universal testing machine (UTM) with a 100 kN capacity, in accordance with the ASTM D790-02 standard. The crosshead speed was set to 5 mm/min, and the base width was 100 mm. The specimens used for testing had dimensions of 191 × 20 × 2 mm^3^. To determine the flexural strength, the following equation was employed:$$\sigma =\frac{3\mathrm{FL}}{{2\mathrm{bt}}^{2}}$$
where σ is flexural strength, F is the maximum load, L is base width, b is width of the specimen, and t is the thickness of the specimen [43].

### 2.2. Factors Affecting Flexural Strength

The objective of the study was to investigate the effect of varying manufacturing parameters on the mechanical properties of composite materials. Various factors, such as ply orientation, number of fabric layers, and infill, have been shown to impact flexural strength in previous research. Nonetheless, this study investigates the effect of a particular variable on flexural strength. The research demonstrates that the calculation of flexural strength is a crucial variable that has a direct effect on the thickness and width values derived from the flexural strength equation. Once the manufacturing of the carbon fiber/epoxy composite with graphite filler is completed, the specimen will be tested for flexural strength using the 3-point bending test, and the data collected will include the flexural strength values. In finding factors that affect flexural strength among the many methods of production, this research conducted studies related to factors and predictions. Chong S.S et al. [44] studied the intensity of color by using the method of finding an equation that can calculate or predict color intensity by various methods. Multiple linear regression is another way to find color intensity, by choosing from factors that affect the previous variables studied. This method will facilitate the identification of factors that impact flexural strength. This research will collect flexural strength data while varying width, thickness, ply orientation, manufacturing parameters, and weight percentage of graphite filler. These obtained values will be analyzed using the IBM^®^ SPSS^®^ Statistics 22.0 program, specifically employing the linear regression method. The researcher has observed a linear trend in the data, indicating the suitability of the linear regression method for prediction purposes. The reliability of the prediction results further supports the selection of this method for the study [45].

### 2.3. Normalization

Normalization involves transforming or scaling existing input data to a common range to ensure equal importance and avoid biases in the training process. In the case of the 103 datasets, normalization was applied as a preprocessing step. Initially, the R-squared (R^2^) value was used as a measure of accuracy, which provided a reliable criterion. However, it was observed that the mean squared error (MSE) value was high, likely due to data anomalies and variations in the range of parameter values. To address this issue, training was conducted using a low learning rate to ensure better convergence and improved performance during the normalization process. To address the mentioned issue, the normalization method was employed to mitigate the problem of high MSE values and ensure that the R^2^ value approaches 1, indicating accurate and reliable data. In this context, the boundary set for normalization was defined as 0 to 1 across the 103 datasets. By applying various algorithms to these datasets, the one yielding the best prediction performance for flexural strength was selected. This chosen algorithm was then utilized for the 487 datasets, encompassing the entire dataset. For these 487 datasets, a length normalization range of −1 to 1 was set. The entire process was verified and implemented using the MATLAB^®^ R2022a program.

### 2.4. Evaluation of Prediction Errors

This research requires the selection of an algorithm for prediction by considering the accuracy from MSE and R^2^. After careful consideration, the LM and SCG algorithms of the ANN with MATLAB^®^ R2022a program were selected. MSE and R^2^ are selected for training, with the formulas as follows:$$\mathrm{MSE}=\sum_{i=1}^{N}\frac{{\left(y_{i}-{\hat{y}}_{i}\right)}^{2}}{N}$$
$$R^{2}=1-\frac{\sum {\left(y_{i}-{\hat{y}}_{i}\right)}^{2}}{\sum {\left(y_{i}-\bar{y}\right)}^{2}}$$
where *N* is the number of data, $y_{i}$ is true value at data *i*, ${\hat{y}}_{i}$ is prediction value at data *i*, and $\bar{y}$ is an average value.

### 2.5. Hidden Layers

In this research, the determination of the optimal number of hidden layers was conducted using the MATLAB^®^ R2022a program. The nntool function was utilized to iterate through different configurations of hidden layers and assess their impact on prediction accuracy. The process involved adjusting the number of hidden layers within the range of 1 to 10, with each hidden layer containing 10 neurons. The criterion for selecting the best configuration was based on the adjustment of the mean squared error (MSE) and R^2^ values. The boundaries of the hyperparameters were defined in accordance with Table 2 (applied to the 103 datasets) and Table 3 (applied to the 487 datasets).

## 3. Results

### 3.1. Analyzing the Factors Affecting Flexural Strength

In analyzing the data, the input and dependent variables must be selected first, in order to analyze the data by the IBM^®^ SPSS^®^ Statistics 22.0 program. In this research, the initial variables are width, thickness, ply orientation, manufacturing, and the percentage of graphite filler (wt% of graphite), while the dependent variable is the flexural strength. The accuracy of the equation for predicting the flexural strength from the initial variables is determined by the R^2^ value. The criteria value for determining the factors that affect the dependent variable is set to 0.05, indicating that the variable being studied affects 95% of the dependent variable. If the significance value from the analysis results is less than 0.05, the predetermined variable will be considered a factor that significantly affects the flexural strength. Table 4 shows that the thickness and ply orientation have the greatest effect on flexural strength, while manufacturing and percentage of graphite filler are minor factors that affect flexural strength.

Based on the analysis conducted using the IBM^®^ SPSS^®^ Statistics 22.0 program to determine the factors influencing flexural strength, this research identified the following input variables that significantly impact flexural strength: thickness, width, ply orientation, manufacturing, and wt% graphite filler. These variables were found to have a substantial influence on the flexural strength of the material.

### 3.2. The Algorithm to Predict Flexural Strength

In this section, a suitable algorithm is discussed for predicting the flexural strength of carbon fiber/epoxy composites to ensure accurate results. To predict the flexural strength, this research utilized five input variables: percentage of graphite filler, manufacturing process, ply orientation, width, and thickness. The output variable was the flexural strength itself. To achieve this, the research employed the Levenberg–Marquardt backpropagation (LMBP) and scaled conjugate gradient (SCG) algorithms of the artificial neural network (ANN) implemented in the MATLAB^®^ R2022a program. The ANN architecture consisted of seven hidden layers (specifically, an architecture of 5-7-1-1), as shown in Table 5. This number of hidden layers was determined as the optimal choice based on the analysis of 103 flexural strength datasets. The dataset considered in this study consists of carbon fiber/epoxy composite testing data at eight layers, resulting in a total of 103 datasets. These datasets were divided into training dataset (70%, 73 datasets), validation dataset (15%, 15 datasets), and testing dataset (15%, 15 datasets). The data within each dataset were randomly selected to ensure the representative nature of the samples used in this study. By choosing these dataset boundaries, the predictions can be made with higher accuracy, as the training data are shared with the validation data, leading to improved data accuracy. The artificial neural network (ANN) parameters were set according to the specifications shown in Table 2. Additionally, Figure 3 illustrates the procedure and steps involved in utilizing the artificial neural network prediction methods.

Using the algorithms of LMBP and SCG, the flexural strength values of both algorithms were compared with the experimental values, as shown in Figure 4. It is found that both algorithms can predict the flexural strength close to the experimental values. To be able to indicate which of the two algorithms can predict flexural strength closer to the results of the experiment, the MSE and R^2^ of the testing data are considered, as shown in Table 6. Prediction using the LM algorithm can predict the flexural strength more accurately than the SCG algorithm. The MSE value of the LM algorithm from the testing data was 0.0044, which was less than the prediction by the SCG algorithm, which was 0.00838, indicating that the LM algorithm’s flexural strength prediction error value demonstrated less prediction error than the SCG algorithm. The R^2^ value from testing with the LM algorithm was 0.9926, which was closer to 1 than the prediction from the SCG algorithm, with an R^2^ of 0.9791, indicating that the LM algorithm was more suitable for predicting the flexural strength than the SCG algorithm.

### 3.3. The Appropriate Number of Hidden Layers for the Data

To predict the flexural strength of the 487 tested specimens, an equal number of datasets, i.e., 487 datasets, will be used with the MATLAB^®^ R2022a program. It is important to note that when the number of datasets changes, normalization and optimization of the number of hidden layers may be necessary to ensure that the MSE and R^2^ values fall within the desired boundaries. This adjustment is required because the data values must be appropriately scaled to fit the program’s requirements. After the program makes predictions based on the normalized data, it is necessary to convert the data back to its original form in order to interpret the predicted values from the artificial neural network (ANN) with the LM algorithm. The hyperparameters of each model were optimized using the LM algorithm during training. The specific data to be considered in this process are presented in Table 3.

The databases were randomly divided into three sets: training data, validation data, and test data, comprising 70%, 15%, and 15% of the total data, respectively. The training data are used to develop and optimize the predictive model, while the test data are the final dataset used to evaluate the model’s performance. In order to determine which model can best predict the flexural strength, an exploration of the optimal number of hidden layers is conducted. The learning rate is adjusted within the range of 0.1 to 0.01, decreasing by increments of 0.01. It was observed that a learning rate of 0.01 yielded the highest prediction accuracy. By selecting 1000 epochs, the researcher aims to set a wide range of epochs to allow for convergence and capture the complexity of the data. Common statistical performance metrics such as mean squared error (MSE) and R-squared (R^2^) are utilized to measure the predictive performance and assess the improvement in model performance [46].

Finding the optimal quantity of hidden layer datasets is required [47]. In this section, considering the number of hidden layers from 1 to 10, the appropriate number of hidden layers for the 487 datasets used to predict flexural strength is shown in Table 7. The suitable number of layers for the dataset used to predict flexural strength was found to be 5-10-1-1, which includes 5 input layers, 10 hidden layers, 1 output layer, and 1 bias layer within the artificial neural network (ANN) architecture, as shown in Figure 5. This prediction demonstrates an MSE value of less than 0.01, and the R^2^ value of training, validation, and testing is greater than or equal to 0.95. This ensures that the error value of flexural strength prediction does not exceed the specified boundary.

The convergence of data, taking into account the error or MSE values, can be observed in Figure 6, considering the 5-10-1-1 prediction structure. The data initially started with higher error values and gradually adjusted to reduce the errors, resulting in more accurate predictions. As depicted in the figure, all three sets of data (training, validation, and testing) converge towards the line representing the lowest error reduction at epoch 94. This indicates that the model achieved optimal performance in terms of minimizing errors and producing reliable predictions.

To evaluate the accuracy of predicting the output data (flexural strength), the R^2^ value was considered, which is represented in the regression plot. Figure 7 displays the R^2^ values for the training, validation, testing, and all data sections. The scatter plot in the Figure 7 shows that the data points (represented by circle symbols) cluster around the fit line or regression line, indicating a strong tendency for the predicted data to align closely with the true values. The closer the fit curve is to the dotted line or the Y = T line, the lower the predictive error, indicating a suitable dataset for prediction purposes.

### 3.4. Comparison of Predicted Values with Experimental Values

The flexural strength prediction from the ANN with the LM algorithm compared with the experimental results is shown in Figure 8. It was found that the flexural strength predicted using the LM algorithm was quite close to the flexural strength obtained from the experimental results, and MSE and R^2^ values of all predicted data were 0.003039 and 0.95274, respectively. This shows that the prediction of flexural strength of carbon fiber/epoxy composites using the ANN method results in high accuracy.

Using the ANN method with the LM algorithm and five input data (percentage of graphite filler, manufacturing, ply orientation, width, and thickness), the flexural strength was predicted. The MSE values of testing and all data were found to be 0.0039 and 0.003039, respectively, which were within the set boundary of the MSE value being lower than 0.001, indicating a low error value and highly accurate prediction. The R^2^ values of testing and all data were 0.95 and 0.95274, respectively, close to 1, which further confirms the high accuracy of the predicted data using this algorithm. In general, the sample size determines the number of datasets required for precise and accurate predictions. This research designates a case study, as shown in Table 1, which provides a comprehensive overview of the investigation. The results reveal that the MSE is less than 0.001 and R^2^ is greater than or equal to 0.95, indicating a high level of prediction accuracy and a sufficient number of samples for this study.

### 3.5. Prediction the Flexural Strength

By comparing the flexural strength predicted by the ANN method with the flexural strength obtained from the experimental results, the accuracy of this model can be determined. Therefore, it can be applied in practice to predict the flexural strength when the data are not in the modeled dataset. To test the accuracy of the model, three sets of predictive data were considered, as shown in Table 8, which includes the input data for carbon fiber/epoxy composite manufacturing.

In order to predict the flexural strength, the data to be predicted must be normalized to the available data to obtain a boundary value between −1 and 1, according to the boundary of the normalized flexural strength. To obtain the predicted value by the flexural strength prediction results of the data in Table 7, the prediction is shown in Table 9.

The prediction result of this model is close to the experimental result, with the MSE value being less than 0.01, indicating that the error in the prediction of flexural strength is small, and R^2^ is more than 0.95, indicating a high prediction accuracy. Therefore, this research can be applied to predict flexural strength without wasting materials, while shortening the testing time.

From the predictions, flexural strength should be compared with other studies to test whether the prediction model can predict the flexural strength of other works. The tendency for accuracy in determining flexural strength verifies the reliability of the manufacturing in this research. This result showcases the reliability of the manufacturing process employed in this research, which is comparable to the test specimens used in other studies. Therefore, the prediction results of flexural strength were compared in this research. The current prediction of flexural strength and values from other studies are shown in Table 10.

It can be seen that the flexural strength of the ANN prediction method and the experimental results are similar. Similarly, when comparing the flexural strength with other studies, it was found to be close to the prediction and value from this study. Therefore, it can be concluded that the flexural strength prediction model of this research can be used to predict values in other research, to the extent specified by the research. By adhering to the specified input parameters, researchers can ensure that the manufactured material meets the desired requirements. This enables the prediction of flexural strength without the need for time-consuming and resource-intensive testing. By accurately predicting the flexural strength, valuable time and resources can be saved in the material manufacturing process.

## 4. Conclusions

The objective of this research is to predict the flexural strength of carbon fiber/epoxy composites. A dataset comprising 487 carbon fiber/epoxy composites was fabricated and used for analysis. The input parameters considered in the study are ply orientation, manufacturing process, percentage of graphite fillers, width, and thickness. The output parameter of interest is the flexural strength. To accomplish the prediction task, this research employed artificial neural network (ANN) methods within the MATLAB^®^ R2022a program. The details of the research methodology and findings are summarized as follows.

1. When taking the values from the 3-point bending test of carbon fiber/epoxy composite to determine the factors that effect to the flexural strength values, it can be seen that the ply orientation, thickness, manufacturing, and percentage of graphite filler significantly affect the flexural strength.

2. Algorithms that can accurately predict flexural strength with input data within the scope specified by this research are the LMBP and SCG algorithms. They have high prediction accuracy and reduced prediction error, which are determined from the MSE and R^2^ values, with an MSE greater than 0.01 and R^2^ greater than or equal to 0.95.

3. To predict the flexural strength of carbon fiber/epoxy composites in various cases, a total of 487 datasets were generated and fed into the learning program using the artificial neural network (ANN) method. The Levenberg–Marquardt backpropagation (LMBP) algorithm was employed, and the results indicated that the prediction of flexural strength achieved mean squared error (MSE) values of 0.0039 for the testing dataset and 0.003039 for all the data. Furthermore, the coefficient of determination (R^2^) was assessed for both the testing dataset and all the data, yielding values of 0.95 and 0.95274, respectively. These R^2^ values align closely with the predefined boundaries, demonstrating the accuracy and effectiveness of the prediction model.

In order to predict the flexural strength based on this research, the input and output data specified within the research scope are the key requirements. The input data comprise ply orientation, manufacturing method, graphite filler percentage, breadth, and thickness. On the other hand, the output data must consist of the flexural strength, as it serves as the basis for implementing the prediction model in this study. Moving forward, the researcher intends to broaden the range of inputs used to predict flexural strength across various manufacturing applications. This expansion will involve the development of more accurate predictions, not only for flexural strength, but also for other vital mechanical properties such as tensile strength and impact strength. By incorporating these enhancements, this research aims to provide a comprehensive understanding of the material’s performance in diverse scenarios.

## Figures and Tables

**Figure 1 materials-16-05301-f001:**
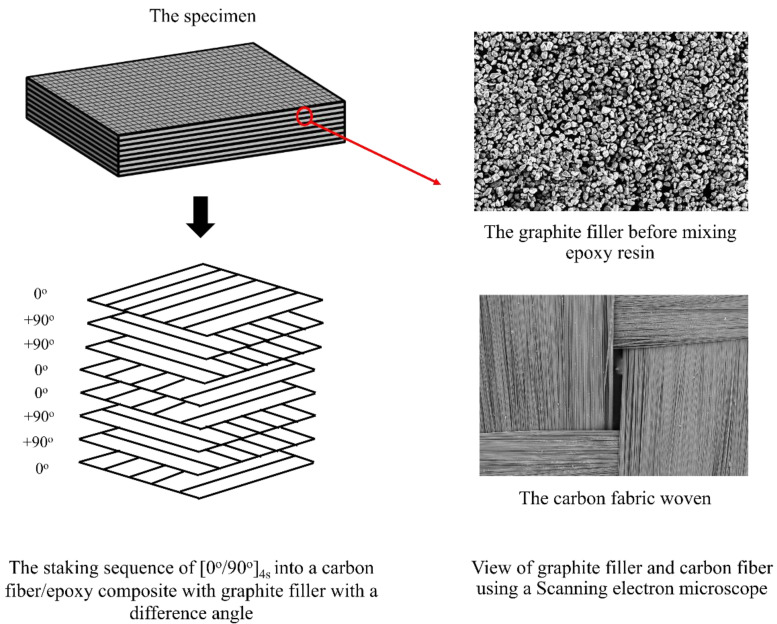
The example specimens with eight layers of carbon fiber fabric (The red circle with red arrows show the epoxy with graphite filler on the specimen).

**Figure 2 materials-16-05301-f002:**
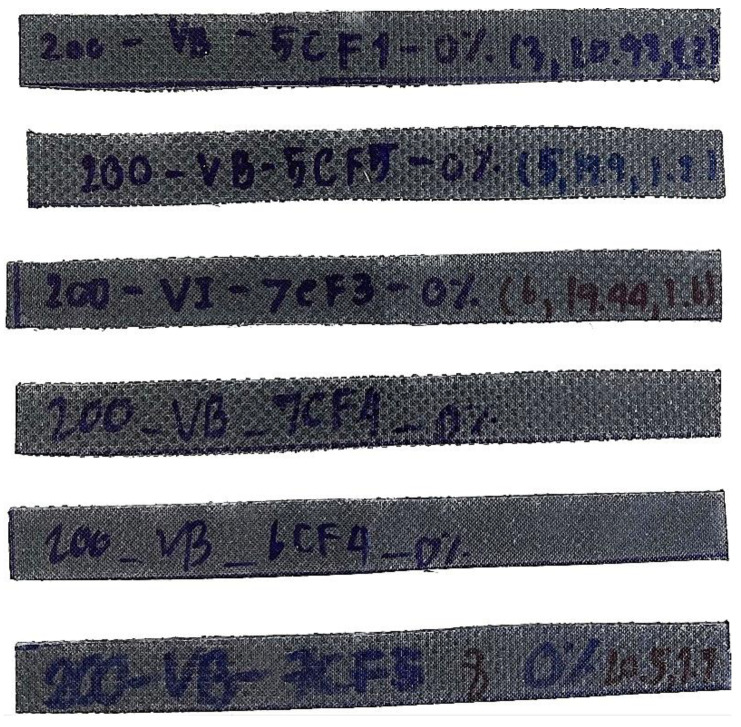
The specimen for 3-point bonding test.

**Figure 3 materials-16-05301-f003:**
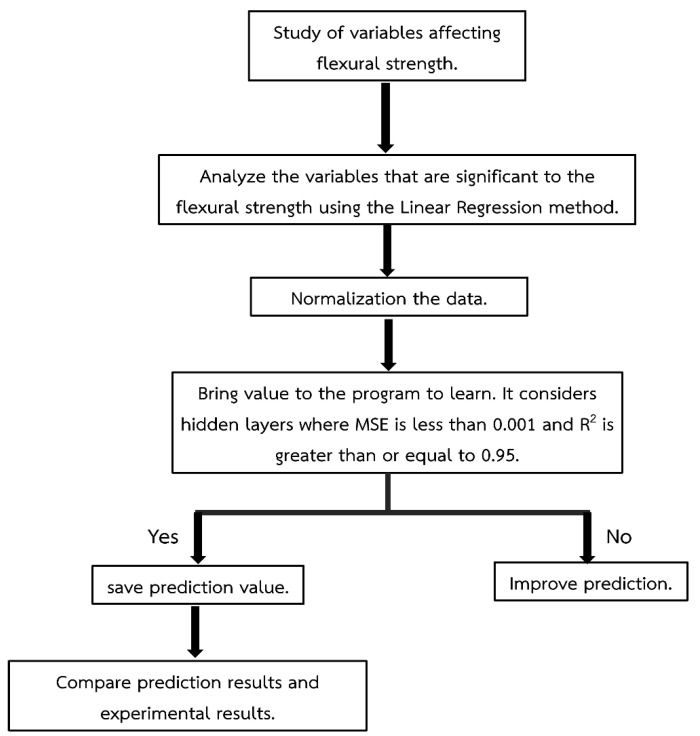
Methods for predicting with an artificial neural network.

**Figure 4 materials-16-05301-f004:**
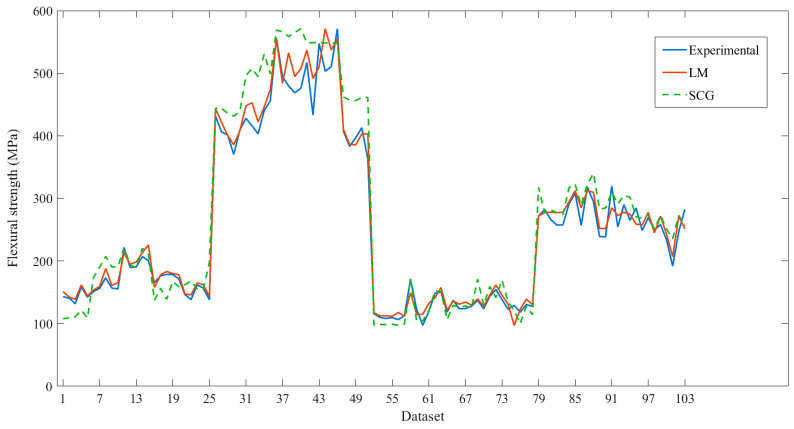
Comparison between experimental and prediction values from the SCG and LM algorithms.

**Figure 5 materials-16-05301-f005:**
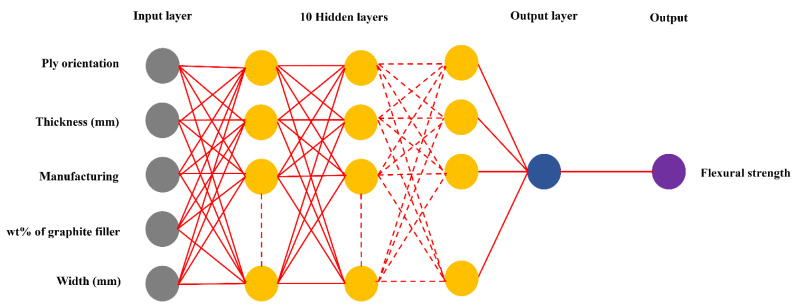
Artificial neural network (ANN) architecture (Circles are neurons in each layer).

**Figure 6 materials-16-05301-f006:**
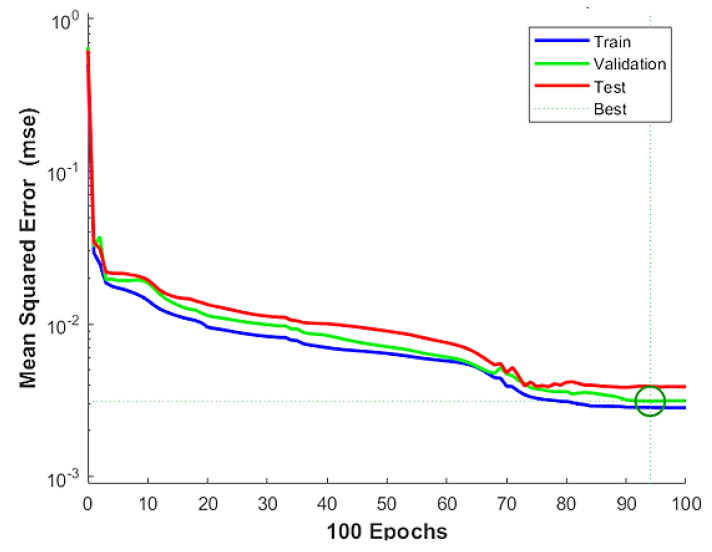
Performance of data from MSE (green color circle is shown the convergence).

**Figure 7 materials-16-05301-f007:**
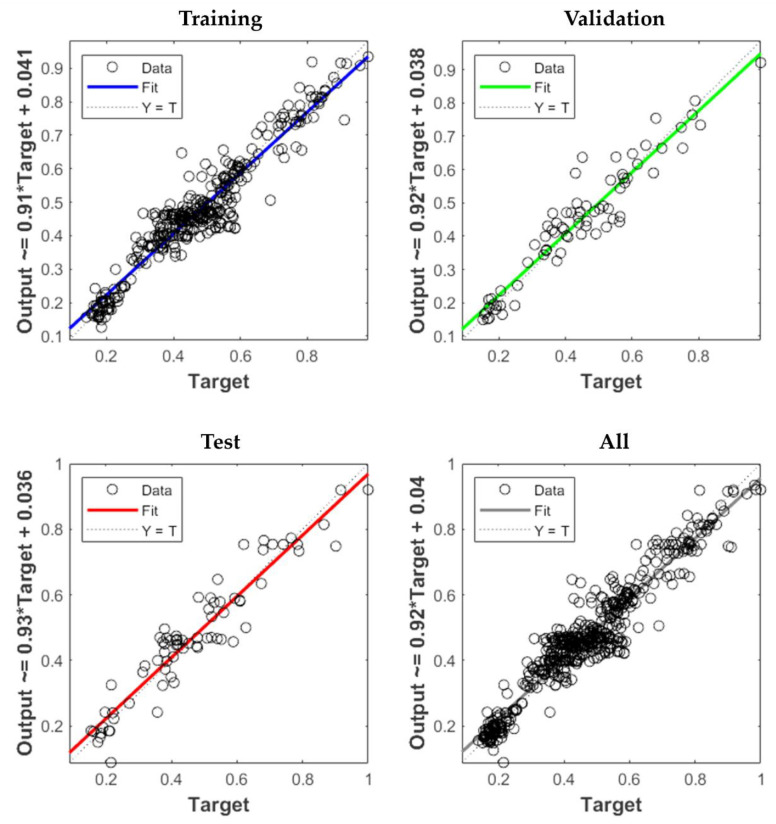
Regression plot of ANN prediction.

**Figure 8 materials-16-05301-f008:**
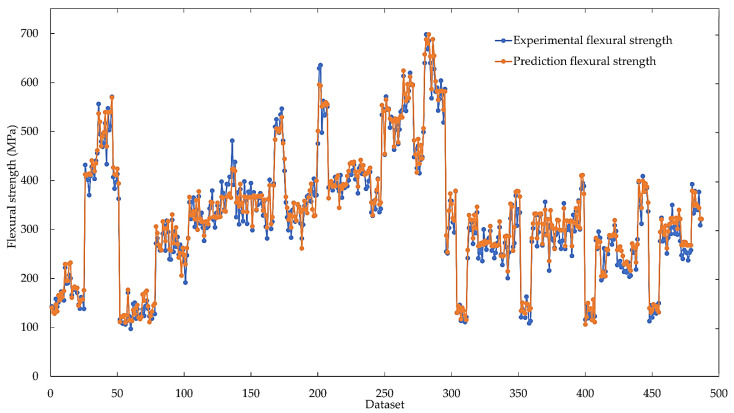
Comparison of the flexural strength of carbon fiber/epoxy composites predicted using the LM algorithm with experimental results.

**Table 1 materials-16-05301-t001:** Condition of manufacturing of carbon fiber/epoxy composite.

Number of Layers	Ply Orientation	Manufacturing	Graphite Filler (wt%)	Number of Specimens
4	[0°/90°]_2s_	Vacuum bagging and Vacuum infusion	0	16
	[45°/−45°]_2s_			16
	[0°/45°]_2s_			16
	[90°/45°]_2s_			16
	[0°/−45°]_2s_			16
	[90°/−45°]_2s_			16
5	[0°/90°/0°/90°/0°]			16
	[45°/−45°/45°/−45°/45°]			16
	[0°/45°/0°/45°/0°]			16
	[90°/45°/90°/45°/90°]			16
	[0°/−45°/0°/−45°/0°]			16
	[90°/−45°/90°/−45°/90°]			16
6	[0°/90°]_3s_			16
	[45°/−45°]_3s_			16
	[0°/45°]_3s_			16
	[90°/45°]_3s_			16
	[0°/−45°]_3s_			16
	[90°/−45°]_3s_			16
7	[0°/90°/0°/90°/0°/90°/0°]			16
	[45°/−45°/45°/−45°/45°/−45°/45°]			16
	[0°/45°/0°/45°/0°/45°/0°]			16
	[90°/45°/90°/45°/90°/45°/90°]			16
	[0°/−45°/0°/−45°/0°/−45°/0°]			16
	[90°/−45°/90°/−45°/90°/−45°/90°]			16
8	[0°/90°]_4s_	Vacuum bagging and Vacuum infusion	0	11
			5	11
			7.5	10
			10	10
			12.5	10
	[−45°/45°]_4s_	Vacuum bagging and Vacuum infusion	0	10
			5	10
			7.5	11
			10	10
			12.5	10

**Table 2 materials-16-05301-t002:** Setting parameters of ANN.

Parameters	Description
Number of input layer	1
Number of output layer	1
Number of hidden layers	7
Learning rate	0.01
Activation function	Sigmoid
Number of epochs	1000
Algorithm	LM and SCG
The number of hidden neurons	10
Number of datasets	103
Training dataset	70% (73 data)
Validation dataset	15% (15 data)
Testing dataset	15% (15 data)
Error function	MSE
Initializer	Random

**Table 3 materials-16-05301-t003:** ANN parameters of carbon fiber/epoxy composite.

Parameters	Description
Number of input layer	1
Number of output layer	1
Number of hidden layers	1–10
Learning rate	0.01
Activation function	Sigmoid
Number of epochs	1000
Algorithm	LM
The number of hidden neurons	10
Number of datasets	487
Training dataset	70% (341 data)
Validation dataset	15% (73 data)
Testing dataset	15% (73 data)
Error function	MSE
Initializer	Random

**Table 4 materials-16-05301-t004:** Factors affecting flexural strength.

Coefficient ^a^		
Model	Significant	Meaning
Thickness	0.000	It greatly affects the flexural strength
Width	0.240	Affects the flexural strength
Ply orientation	0.000	It greatly affects the flexural strength
Manufacturing	0.001	It greatly affects the flexural strength
wt% of graphite	0.001	It greatly affects the flexural strength

^a^. Dependent Variable Flexural strength.

**Table 5 materials-16-05301-t005:** Comparison of MSE and R^2^ values to find the optimal number of hidden layers from 103 datasets.

Architecture	Testing Set MSE	Training Set MSE	Validation Set MSE
5-1-1-1	2.61 × 10^−2^	1.37 × 10^−2^	1.05 × 10^−2^
5-2-1-1	8.92 × 10^−3^	1.01 × 10^−2^	1.88 × 10^−2^
5-3-1-1	1.53 × 10^−2^	8.73 × 10^−3^	5.06 × 10^−3^
5-4-1-1	8.96 × 10^−3^	3.79 × 10^−3^	1.13 × 10^−2^
5-5-1-1	1.11 × 10^−2^	2.14 × 10^−2^	9.81 × 10^−3^
5-6-1-1	5.78 × 10^−3^	2.79 × 10^−3^	3.75 × 10^−3^
5-7-1-1 ^A^	4.4 × 10^−3^	2.08 × 10^−3^	2.41 × 10^−3^
5-8-1-1	5.22 × 10^−3^	3.54 × 10^−3^	1.63 × 10^−2^
5-9-1-1	1.40 × 10^−2^	2.80 × 10^−3^	7.89 × 10^−3^
5-10-1-1	1.61 × 10^−2^	1.14 × 10^−2^	1.07 × 10^−2^

^A^ The selected ANN network structure.

**Table 6 materials-16-05301-t006:** Appropriate algorithms were used to predict flexural strength.

Method	R^2^		MSE	
	Test	Overall	Test	Overall
LM ^b^	0.9926	0.9925	4.4 × 10^−3^	4.3 × 10^−3^
SCG	0.9791	0.9823	8.38 × 10^−3^	9.06 × 10^−3^

^b^ LM is a suitable method for predicting flexural strength more than SCG.

**Table 7 materials-16-05301-t007:** Comparison of MSE and R^2^ values to find the optimal number of hidden layers from 487 datasets.

Network Structure	Training MSE		Validation		Testing	
	MSE	R^2^	MSE	R^2^	MSE	R^2^
5-1-1-1	2.24 × 10^−2^	0.57	2.67 × 10^−2^	0.39	2.40 × 10^−2^	0.55
5-2-1-1	1.25 × 10^−2^	0.79	1.15 × 10^−2^	0.79	1.86 × 10^−2^	0.70
5-3-1-1	1.36 × 10^−2^	0.76	1.05 × 10^−2^	0.81	1.09 × 10^−2^	0.85
5-4-1-1	1.05 × 10^−2^	0.83	1.10 × 10^−2^	0.82	1.16 × 10^−2^	0.79
5-5-1-1	9.13 × 10^−3^	0.83	1.45 × 10^−2^	0.82	1.17 × 10^−2^	0.84
5-6-1-1	5.88 × 10^−3^	0.90	9.33 × 10^−3^	0.86	1.06 × 10^−3^	0.84
5-7-1-1	3.95 × 10^−3^	0.93	7.02 × 10^−3^	0.93	4.90 × 10^−3^	0.93
5-8-1-1	5.47 × 10^−3^	0.91	6.60 × 10^−3^	0.91	9.70 × 10^−3^	0.85
5-9-1-1	3.29 × 10^−3^	0.95	3.91 × 10^−3^	0.94	3.83 × 10^−3^	0.94
5-10-1-1 ^c^	2.84 × 10^−3^	0.95	3.12 × 10^−3^	0.95	3.90 × 10^−3^	0.95

^c^ The selected ANN network structure.

**Table 8 materials-16-05301-t008:** Carbon fiber/epoxy composite manufacturing data or predictive input.

Dataset	Thickness (mm)	Width (mm)	Ply Orientation	Manufacturing	%wt Graphite Filler (wt%)
1	2.1	23	[−45 °/45°]_4s_	Vacuum bagging	12.5
2	2	17	[0°/90°]_4s_	Vacuum bagging	10
3	2.4	20.2	[−45°/45°]_4s_	Vacuum infusion	7.5
4	0.9	18	[0°/−45°]_s_	Vacuum bagging	5
5	1.63	19.8	[0°/90°/0°/90°/0°/90°/0°]	Vacuum infusion	0
6	1.2	20.6	[0°/45°/0°/45°/0°]	Vacuum infusion	0
7	0.8	20.5	[0°/90°]_3s_	Vacuum bagging	0
8	1.1	20.34	[90°/−45°/90°/−45°/90°]	Vacuum infusion	0

**Table 9 materials-16-05301-t009:** Comparison data of flexural strength between prediction and experimental results.

Dataset	Flexural Strength from Prediction (MPa)	Flexural Strength from Experimental (MPa)	MSE	R^2^
1	159.8931	152.5143	2.33 × 10^−10^	0.9977
2	394.8466	384.621	4.38 × 10^−7^	0.9797
3	317.87	296.05	2.03 × 10^−7^	0.9558
4	128.1012	122.5136	8.49 × 10^−10^	0.9999
5	679.3791	668.4299	4.99 × 10^−7^	0.9900
6	330.0794	327.7892	6.41 × 10^−10^	0.9999
7	382.3560	358.2715	1.07 × 10^−6^	0.9698
8	347.8172	339.9457	3.46 × 10^−7^	0.9639

**Table 10 materials-16-05301-t010:** Comparison of flexural strength in this research with other studies.

Specimens	Flexural Strength from Researcher (MPa)	Flexural Strength from Prediction (MPa) (Present Work)	Flexural Strength from Experimental (MPa) (Present Work)
[−45°/45°]_4s_	105.49 [48]	92.78	96.13
[0°/90°/0°/90°/0°]	754 [49]	648.5	635.89
[45°/−45°/45°]_s_	167.39 [50]	135.01	149.06

## Data Availability

Not applicable.

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
