# Peer review of "The Flexural Strength Prediction of Carbon Fiber/Epoxy Composite Using Artificial Neural Network Approach"

_materials, 2023, doi:10.3390/ma16155301_

Round 1
Reviewer 1 Report
The current paper uses experimental and artificial intelligence methods to predict the flexural strength of carbon fiber composites. Some results obtained are innovative and can provide the readers for performance evaluating and predicting methods of composites. However, the authors are encouraged to consider the following comments to make necessary improvements.
1. The current abstract does not convey any key information, and the research background and significance are not clear. It is recommended that the authors modify the current abstract through providing some key results and conclusions.
2.Introduction, the following comments should be further responded.
(1) Why does the use of natural materials have a negative impact on the environment? Generally speaking, natural materials have less environmental pollution compared to synthetic materials.
(2) For carbon fiber reinforced polymer composites, their properties and advantages should be emphasized compared to other fiber composites, such as higher mechanical properties, excellent fatigue resistance, corrosion resistance, creep resistance, etc. On the other hand, the relatively high price of carbon fiber should also be considered in terms of cost performance in engineering applications. Please review the relevant research below and make necessary additions. Composite Structures, 2022, 293, 115719. Composites Part B: Engineering, 2022, 241:110020. Mechanics of Advanced Materials and Structures, 2023, 30(4):814-834.
3. In Part 2, the authors observed the fiber distributions by scanning electron microscopy. It is recommended to provide the testing method for scanning electron microscopy.
4. The color of image 2 is too dim, it is recommended to increase the brightness.
5. Please explain why this article focuses on the flexural strength of materials, and generally speaking, the tensile strength of CFRP should be given more consideration in practical engineering applications.
6. The process and steps of using artificial neural network prediction methods should be described in detail.
7. Why are the main parameters such as thickness and width selected in Table 2? Are these parameters typically representative?
8. The number of the figure is incorrect. It is recommended to adjust or modify it. Additionally, there are extra right and upper scale value in the figure, and it is recommended to remove them. In addition, the clarity of all images should be further improved.
9. Is the amount of data used in the current paper sufficient to predict the bending strength of materials? How many sets of data are generally required to ensure the accuracy and accuracy of predictions?
10. Can the current model expand the data prediction from other research work? How to comprehensively evaluate the prediction stability of the model?
Reviewer 2 Report
In this work, An artificial neural network method was used to study the flexural strength of carbon fiber/epoxy composites. The input parameters considered were ply orientation, manufacturing, width, thickness, and percentage of graphite filler. The output parameter was flexural strength.This work is very interesting, and in my opinion, it can be accepted after modification. The comments are as follows:
1. Manufacturing process, as a factor mentioned in this paper, has no persuasive experimental results to prove it.
2. Please check the manuscript for grammatical and formal errors.
3. The description of the experimental process of relevant materials needs to be supplemented and improved. Although this paper focuses on software analysis, it also needs to be analyzed from the perspective of materials.
4. Too few sets of prediction group and experimental group, and too few sets of control group are not very persuasive.
In this work, An artificial neural network method was used to study the flexural strength of carbon fiber/epoxy composites. The input parameters considered were ply orientation, manufacturing, width, thickness, and percentage of graphite filler. The output parameter was flexural strength.This work is very interesting, and in my opinion, it can be accepted after modification. The comments are as follows:
1. Manufacturing process, as a factor mentioned in this paper, has no persuasive experimental results to prove it.
2. Please check the manuscript for grammatical and formal errors.
3. The description of the experimental process of relevant materials needs to be supplemented and improved. Although this paper focuses on software analysis, it also needs to be analyzed from the perspective of materials.
4. Too few sets of prediction group and experimental group, and too few sets of control group are not very persuasive.
Reviewer 3 Report
Please, refer to the attached document.

The paper requires careful proofreading to correct several inaccuracies, as commented in the attached document.
Round 2
Reviewer 1 Report
The authors have well responded to all comments from the reviewers. Another small issue is to check the reference to ensure that the format of the references is consistent.
Good.
Reviewer 3 Report
Please, refer to the attached comments.

Teh paper contains many inaccurate sentencdes and writings, as commented in the attached doccument.
Round 3
Reviewer 3 Report
This paper continues to contain significant shortcomings, even after the second revision. The authors have failed to address the feedback given in the prior comments adequately, and a new critical error has been identified in this version of the manuscript. The authors suggest, in Lines 304-306, that the architecture of the Artificial Neural Network (ANN) model employed in this study, specifically the number of hidden layers, is borrowed from a separate investigation. This approach is not acceptable. The ANN model's structure, including all hyperparameters, such as the number of hidden layers, activation function, number of hidden neurons, and other essential hyperparameters, should be specifically optimized for this study based on the data used in this study.
Moreover, the article referenced by the authors for this approach is neither accessible nor published, and it covers an entirely different subject; hence the data used differs, as suggested by the title: Saensuriwong, K.; Phunpeng, V.; Uangpairoj, P.; Thammakul, K. Comparative study of Training Algorithm of Artificial Neural Network for Carbon Fiber/Epoxy Composite Materials. AIP 2023, 1-7. (In press).
Given this scenario, the study offers no significant contribution and is rendered completely invalid due to this major flaw. I kindly request the authors to thoroughly review the comments provided and address each point fully and accurately.
The paper contains some inaccuracies in its sentences and writing, as previously commented.
Author Response
The authors of this paper would like to thank to the innovation comments and suggestions from three reviewers.